# Nanostructured ZnO/Ag Film Prepared by Magnetron Sputtering Method for Fast Response of Ammonia Gas Detection

**DOI:** 10.3390/molecules25081899

**Published:** 2020-04-20

**Authors:** Yiran Zheng, Min Li, Xiaoyan Wen, Ho-Pui Ho, Haifei Lu

**Affiliations:** 1School of Science, Wuhan University of Technology, Wuhan 430070, China; 2Department of Biomedical Engineering, The Chinese University of Hong Kong, Shatin, Hong Kong, China

**Keywords:** nanostructured film, ammonia gas sensor, fast response, magnetron sputtering method, semiconductor

## Abstract

Possessing a large surface-to-volume ratio is significant to the sensitive gas detection of semiconductor nanostructures. Here, we propose a fast-response ammonia gas sensor based on porous nanostructured zinc oxide (ZnO) film, which is fabricated through physical vapor deposition and subsequent thermal annealing. In general, an extremely thin silver (Ag) layer (1, 3, 5 nm) and a 100 nm ZnO film are sequentially deposited on the SiO_2_/Si substrate by a magnetron sputtering method. The porous nanostructure of ZnO film is formed after thermal annealing contributed by the diffusion of Ag among ZnO crystal grains and the expansion of the ZnO film. Different thicknesses of the Ag layer help the formation of different sizes and quantities of hollows uniformly distributed in the ZnO film, which is demonstrated to hold superior gas sensing abilities than the compact ZnO film. The responses of the different porous ZnO films were also investigated in the ammonia concentration range of 10 to 300 ppm. Experimental results demonstrate that the ZnO/Ag(3 nm) sensor possesses a good electrical resistance variation of 85.74% after exposing the sample to 300 ppm ammonia gas for 310 s. Interestingly, a fast response of 61.18% in 60 s for 300 ppm ammonia gas has been achieved from the ZnO/Ag(5 nm) sensor, which costs only 6 s for the response increase to 10%. Therefore, this controllable, porous, nanostructured ZnO film maintaining a sensitive gas response, fabricated by the physical deposition approach, will be of great interest to the gas-sensing community.

## 1. Introduction

Liquid ammonia, widely used as a raw material for medicines, pesticides, organic chemical products, and refrigerants, is an important but dangerous chemical. However, improper storage of liquid ammonia may cause leakage of ammonia gas, posing a great threat to people’s lives and property. Gas ammonia (NH_3_) is a toxic, explosive, and colorless gas with a pungent, suffocating odor. In a short time, the upper limit exposure concentration of ammonia for healthy humans is 96 ppm. Otherwise, ammonia can severely irritate human respiratory organs, skin, and eyes [1]. The long-term permissible exposure limit of ammonia in indoor areas is therefore set at 25 ppm. The immediately dangerous to life or health (IDLH) concentration of ammonia is 300 ppm [2]. While research into ammonia gas detection has made great progress, the requirements for the sensing performance of gas sensors are gradually increasing. Hence, in pursuit of improving the gas response of ammonia sensors, the needs for rapid responses to ammonia gas are attracting more and more attention.

Zinc oxide (ZnO) is one of the best potential candidates for the detection of ammonia gas, due to its attractive and distinctive properties [3,4]. The use of ZnO gas sensors applied to ammonia detection has been studied in different ways. In recent years, these studies are mainly in the pursuit of a large surface-to-volume ratio and heterojunction interface, with a simultaneous positive effect [5,6,7,8]. The metal oxides of ZnO have demonstrated to be fabricated with different structures, among which the nanostructured ZnO such as nanoellipsoids [6], nanorods [8], nanoparticles [9], nanoflowers [10], etc. are favorable for the preparation of gas sensors [11] due to the capability of constructing a high surface-to-volume ratio layer [12]. As for the preparation methods, various chemical-reaction-based approaches have been demonstrated to fabricate the nanostructured ZnO layer, such as chemical vapor deposition [13], spray pyrolysis [14], the sol-gel method [15], hydrothermal synthesis [16], etc., while fewer ZnO nanostructures have been demonstrated through the physical fabrication method. To further improve the sensing ability of ZnO, introducing heterojunction with other metal or metal oxide materials is also conventionally proposed [17,18]. For example, Shingange et al. demonstrated that ZnO nanorods were functionalized with Au nanoparticles, yielding a high selectivity towards NH_3_ compared with CO, CH_4_, and H_2_ at the concentration of 100 ppm [8]. Meanwhile, n-ZnO/p-NiO nanofibers fabricated by Lokesh et al. achieved the gas response of 67 for 250 ppm of ammonia because of the n-p heterostructure [19]. The metal or metal oxide materials composited with ZnO are booming [20,21,22,23] and are not listed one by one here. Besides, ZnO formed heterojunctions with other highly conductive materials like rGO [5], SWCNT [24], PPy [25], and PANI [26], helping to reduce the working temperature successfully.

In this present work, an ammonia gas sensor based on nanostructured ZnO film with uniformly distributed nano-hollows is successfully fabricated through physical preparation methods. The porous ZnO film was prepared through successive coating of an extremely thin Ag layer (1, 3, 5 nm) and a 100 nm ZnO film on the SiO_2_/Si substrate through a magnetron sputtering system, which is followed by thermal annealing. The formation mechanism of the porous ZnO film is briefly discussed. The sensing properties of the sensors for ammonia gas are investigated through resistance measurement between the interdigital electrodes on the nanostructured ZnO film. Besides the improved sensitivity over the compact ZnO film, the fast response of the different porous ZnO films were additionally compared and studied under the concentration range of 10 to 300 ppm at an operating temperature of 300 °C. Our proposed physically deposited porous ZnO nanostructured film, which achieves a one-minute warning function, has potential as a mobile ammonia alarm due to its fast response, high stability, and feasibility of miniaturization.

## 2. Experimental Details

### 2.1. Preparation of ZnO Film Sensors

A commercial silicon (100) wafer (1.7 cm × 0.9 cm) with a 270 nm oxide layer on top of its surface was used as the substrate of the sensor. The growth chamber was evacuated to a base pressure of 3.0 × 10^−4^ Pa. Ar gas as a sputtering gas was flown into the chamber to maintain the total working pressure as 2.0 Pa. The ZnO target was pre-sputtered for 5 min to remove surface contaminations. Pure ZnO film and ZnO/Ag nanostructured films were deposited on the substrates respectively at room temperature by the radio frequency (RF) magnetron sputtering system, and the ZnO layers were set to the same thickness of 100 nm. The underlying Ag layer was prepared with different thickness of 1, 3, and 5 nm separately. Then, the samples including pure ZnO film and ZnO/Ag nanostructured films were placed in a muffle furnace and annealed at 750 °C for 1 h in air. Finally, interdigital platinum electrodes were coated onto the samples by a direct current (DC) magnetron sputtering system for the next step of electrical characterization under different gas conditions. The spacing between Pt interdigitated electrodes was 0.25 mm.

### 2.2. Characterizations of ZnO Sensors

The crystal structures of different ZnO films were investigated by using an X-ray diffraction (XRD) instrument (Empyrean produced by PANalytical B.V. in Almelo, the Netherlands). The X-ray source was a copper line focus X-ray diffraction tube operating at 40 kV and 30 mA. Data were collected in the 10.01–89.99° range with 0.0100° steps. The morphologies of the film sensors were characterized using a field emission scanning electron microscope (Zeiss Ultra Plus SEM produced by Carl Zeiss AG in Jena, Germany). The extra high tension (EHT) used in SEM is 5 KV. During the SEM characterization, elemental analysis of the samples was further carried out by energy dispersive X-ray spectroscopy (EDS). EDS completes the map scanning and uses Si Kα1, Zn Lα1,2, O Kα1, and Ag Lα1 to draw the scanned images. X-ray photoelectron spectroscopy (XPS, ESCALAB 250Xi produced by Thermo Fisher Scientific in Waltham, MA, USA) was used to characterize the films for the purpose of identifying the Ag valence state. The X-rays of micro-area XPS analysis were generated by an Al target and by irradiating the samples stuck to the sample stage.

### 2.3. Gas-Sensing Measurement

The ZnO gas-sensing system is shown in Figure 1. The gas mixing chamber or mass flow controller can precisely mix N_2_, O_2_, and NH_3_ gases in proportion. The concentration of NH_3_ filled in the testing chamber is adjusted to the range of 10–333 ppm. A vacuum pump is connected to the other end of the gas testing chamber to keep the outtake speed and the intake speed consistent, and the inside air pressure is kept at 1.01 × 10^5^ Pa. Meanwhile, a heater is adopted to heat the substrate and keep the sensor at 300 °C during sensing characterization, and the background gas is a mixture of N_2_ and O_2_ with a volume ratio of 7:3 before the filling of a specific concentration of NH_3_. The interdigitated platinum electrodes on the surface of the film are connected to the source meter (Keithley 2400) through copper wires. The source meter is controlled by a program in a computer. During the electrical characterization, a fixed voltage of 3 V is applied to the ZnO film, and the current is measured and recorded in real time.

In this study, the gas response (S) is defined as the ratio of the resistance variation of the sensor caused by the target gas to its initial resistance, expressed as
S = |R_a_ − R_g_|/R_a_,(1)
where R_g_ and R_a_ are the resistances of the sensor in NH_3_ gas and air, respectively. According to previous studies, the response or recovery time is generally defined as the duration for the sensors to decrease to 10% or recover to 90% of the initial resistance upon their exposure to the target gas or air. Here, another two parameters, t_90%_ and S_60s_ representing the response duration for the sensor reducing 90% of its initial resistance after its exposure to the target gas and the immediate response of the sensor after a target gas exposure of 60 s, are defined to compare the fast response of the gas sensors.

## 3. Results and Discussion

It is well known that the thermal annealing process is beneficial to the crystallization of metal oxide film in the physical vapor deposition method, whereas relatively less effect on the morphological evolution has been observed [11,27]. As the SEM images in Figure 2a,b show, the pure ZnO film after thermal annealing is compact. Meanwhile, as shown in Figure 2c–h, a great number of nano-sized pores have been observed on the ZnO films pre-coated with different thicknesses of the Ag layer (1, 3, and 5 nm) beneath after thermal annealing, and the size of pores is dependent on the thickness of pre-deposited silver film. To understand the morphological evolution of the nanostructured film, a schematic diagram is shown in Figure 3. Since the ZnO films were deposited on substrates without any heating, the crystallization of the as-prepared film is poor; however, it can be improved under high temperature thermal annealing. During the thermal treatment, re-crystallization of ZnO will happen in the film under the thermal activation of Zn and O atoms. However, it is also difficult to promote the formation of ZnO film with single crystals after thermal annealing. Therefore, multicrystals of the ZnO will be finally obtained, in which boundaries or defects between the regrown ZnO nanocrystals will be present. It should also be mentioned that the spaces among the nanocrystals are so small that the final pure ZnO film remains compact after thermal annealing, as seen in the schematic figure shown in Figure 3a.

However, as shown in Figure 3b, a different morphological evolution of ZnO film will appear when there is a layer of silver, which is highly active under high temperature conditions. To understand the formation of the nanostructured film, the nanolayers of 1 nm, 3 nm, and 5 nm silver deposited on a silicon substrate were characterized with a scanning electron microscope, and the SEM images are shown in Appendix A. As the SEM images show, all silver nanolayers are discontinuous and composed of silver islands. During the thermal annealing of the ZnO/Ag film, besides the recrystallization in the ZnO film, the silver layer underneath the ZnO film will experience deformation and tends to diffuse into the ZnO film [28,29]. Due to the deformation of the planar silver layer a sphere, ZnO crystals above the silver layer will be exposed to extra stress, inducing dislocation among the nanocrystals. Apparently, the grain boundaries or dislocation between the ZnO crystals are the most suitable path for the diffusion of silver atoms, as compared with their insertion into the crystal planes of ZnO [30]. During the diffusion of silver atoms, the boundaries among ZnO nanocrystals will be further expanded, and the nanostructured film with mesopores and macropores inside will finally be formed. The diffused silver atoms will finally condense when the sample is cooling down and will stay around the ZnO nanocrystal surface. It is also reasonable to conclude that the higher the thickness of the silver layer under the ZnO film, the more silver atoms will diffuse through the ZnO film to generate larger pores. That provides us with an effective pathway to control the formation of nanostructured ZnO film.

X-ray diffraction was further used for the characterization of the films, and the experimental results are shown in Figure 4. In the XRD pattern, there are strong diffraction peaks due to the silicon (100) substrate. The key diffraction peaks from ZnO and Ag are marked. As indicated in the XRD patterns, all of the thermally annealed nanostructured films pre-coated with different thicknesses of silver layer show distinguished peaks at 34.49°, which is identical to the standard diffraction of (002) planes of ZnO [29,31]. Meanwhile, the diffraction peak of ZnO (002) is 33.91° for the pure ZnO film after thermal annealing, which shifts to the smaller angle. That can be attributed to the remaining stress between the dense ZnO lattices [32]. In general, the ZnO grains have hexagonal wurtzite-type crystals and are c-axis oriented perpendicular to the substrate. A slight increase of the lattice constant of (002) can be understood as the ZnO grains are compressed along the parallel direction of the substrate by the remaining stress, which also proves that the nanocrystals of ZnO are compact in the pure film. The average crystallite sizes of the ZnO crystal were calculated using the Scherrer equation, as listed in Appendix A. The average crystallite sizes of pure ZnO and Zn/Ag (1, 3, 5 nm) film were 15.45 nm, 29.73 nm, 17.49 nm, and 22.02 nm. Due to the porous nanostructures, no diffraction peak shift was observed for the ZnO films in the presence of silver nanolayers after thermal annealing. A slight decrease of the diffraction intensity from ZnO has been also observed for the nanostructured film as compared to the pure ZnO film, indicating that the diffusion of silver will affect the crystallization of ZnO film. It was also noticed that the diffraction peak of Ag (111) was clearly shown in the XRD pattern of the thermally annealed ZnO film pre-coated with a 5 nm silver layer, revealing the presence of silver nanocrystals in the nanostructured film. However, less or nearly no diffraction was observed for the samples with 3 nm and 1 nm silver layers, which can be ascribed to the relatively lower amount of silver atoms, generating fewer silver nanocrystals to be detected by XRD method.

To further clarify the presence of Ag in the nanostructured films, X-ray photoelectron spectroscopy (XPS) was used for the analysis of the four samples, and the spectrum is shown in Figure 5. In the XPS spectrum, only ZnO/Ag (3, 5 nm) films show a recognizable signal of Ag. The chemical states of AgO, Ag_2_O, and Ag correspond to the binding energies of 367.4 eV, 367.8 eV, and 368.2 eV, respectively [33]. As shown in the spectrum, due to the binding energy of Ag-3d at 368.2° and 373.8°, the chemical state of Ag in the nanostructured films with a 3 nm or 5 nm silver layer can be determined as Ag^0^, which proves that Ag is not oxidized after high temperature thermal annealing. Furthermore, we list the XPS spectrum of the Ag element measured from the annealed pure ZnO and ZnO/Ag (1 nm) films in Appendix A. In order to suppress the noise, we marked the peaks of the XPS spectrum in red. The peaks belonging to Ag 3d5/2 can be observed at 368.2 eV in the spectrum of ZnO/Ag (1 nm), while no distinguished peak of Ag appear in the pure ZnO. As for the reason for no oxidized silver being observed in the XPS spectra, on one hand, silver oxide will be decomposed to silver and oxygen at temperatures of 200 °C or above. On the other hand, it could be due to too little silver oxide in the film after cooling down to be detected through XRD or XPS techniques. Besides, the appearance of signals in other regions is also consistent with the experimental results of other articles [34,35], which is no anomaly. While relatively fewer amounts of silver atoms diffused from the bottom layer, the XPS peak of silver is difficult to distinguish in the annealed nanostructured film pre-coated with a 1 nm silver layer. Furthermore, the distribution of silver on the nanostructured film was investigated by energy dispersive X-ray spectroscopy (EDS). As seen in the experimental result shown in Figure 6, which is characterized from the top and cross-section of annealed nanostructured film pre-coated with 5 nm of silver, uniform distributions of Zn, O, and Ag in the film were observed. That proves that the diffusion of silver atoms from the bottom layer and their final location in the ZnO film.

As shown in Appendix A, it can be observed that the pure ZnO and ZnO/Ag (1 nm) are about kΩ levels, whereas ZnO/Ag (3 nm) and ZnO/Ag (5 nm) are about MΩ levels. The damage to the ZnO film brought by the diffusion of 3 nm or 5 nm Ag is much larger than that brought by 1 nm Ag layer. Therefore, we believe the pre-deposited silver layer is helpful for the formation of nanostructured film during thermal annealing, while contributing less to the conductivity of the nanostructured film. When the substrates are heated to 300 °C, the resistances of all four films are decreased. That can be attributed to the generation of large number of electrons and holes in the ZnO semiconductor under the thermal excitation, which can help to capture more O_2_ on the ZnO surface through donating free electrons.

Owing to the presence of mesopores and macropores in the annealed nanostructured ZnO film increasing the surface-to-volume ratio, the gas-sensing ability of the nanostructured film should be much better than the compact ZnO film. Figure 7 shows the sensing performances of thermal annealed ZnO films with and without Ag layers exposed to a relatively high concentration of 333 ppm NH_3_. As expected, the maximal responses of the three nanostructured films with silver are all better than the gas response of pure ZnO films, as shown in Figure 7a. Besides the enhanced sensitivity, as shown in Figure 7b, the gas-sensing speeds of nanostructured films are also observed to be improved as compared with the compact film, which are characterized by the parameter of t_90%_ as defined in the experimental section. Less time is necessary for the nanostructured films to exhibit 90% resistance variation, which is beneficial to the application of immediate warning of the dangerous gas. Therefore, the sensing ability of the porous nanostructured ZnO films, including the maximal sensitivity and sensing speed, has been clearly demonstrated to be greatly enhanced.

In order to further characterize the gas-sensing properties of the porous nanostructured film, the nanostructured films with 1, 3, and 5 nm silver layers were exposed to 10–300 ppm concentrations of NH_3_. As shown in Figure 8a, the nanostructured films of ZnO/Ag (3 nm) and ZnO/Ag (5 nm) exhibit much better maximum responses than the nanostructured film of ZnO/Ag (1 nm) all through the concentration range of 10–300 ppm. Actually, the maximum response of a sample can reflect the effective surface area for adsorbing the gas molecules. The more cracks formed in the nanostructured film, the larger the achieved surface-to-ratio or effective surface area for capturing the target gas, which consequentially enhances the resistance variation of nanostructured film. However, the pores on the top surface are the major entrances for the diffusion of ammonia into the nanostructure. It is necessary to characterize the transversal property of the pores on the surface. Three typical SEM images of the nanostructured films were selected, and the pores in the images were input into Matlab for statistical analysis. The SEM images and related statistical data of the pores are shown in the Appendix A. As the statistical results show, ZnO/Ag (3, 5 nm) films have a larger pore size and total transversal pore area on the top surface than the sample of ZnO/Ag (1 nm). Therefore, we can conclude that the effective surface areas of ZnO/Ag (3 nm) and ZnO/Ag (5 nm) for ammonia adsorption are much larger than the sample of ZnO/Ag (1 nm), which also can be implied by the SEM image in Figure 2c, exhibiting relatively smaller mesopores in the nanostructured film. Meanwhile, the effective surface area of ZnO/Ag (3 nm) is slightly larger than that of ZnO/Ag (5 nm). Both of the gas sensors from ZnO/Ag (3 nm) and ZnO/Ag (5 nm) showed extraordinarily high response values over a broad concentration range, exhibiting 86.9% and 79.8% at a high NH_3_ concentration of 300 ppm and more than 60% at a low concentration of 10 ppm. That signifies that the two samples are capable of detecting even lower concentrations of gas. It is also important to mention the short-circuiting of the annealed nanostructured films with slightly higher thicknesses of pre-deposited silver layer had been observed. Actually, we prepared Ag layers of 1, 3, 5, 7, and 9 nm underneath the ZnO film. However, we found that the samples with 7 nm of Ag and 9 nm of Ag were short-circuited, which is useless in gas sensing; thus, these have not been characterized here. Therefore, we do not show the morphological and gas sensing characterization results of the nanostructured films of ZnO/Ag (7 nm) and ZnO/Ag (9 nm).

Besides the outstanding parameter of maximum response over different concentrations of ammonia, the time-dependent response is also important in practical applications. Achieving a high response to specific concentrations of gas over a short time is favorable to immediate detection or alarming. Therefore, the two parameters of S_60s_ and t_90%_, defined in the experimental section, are proposed to characterize the fast response of the nanostructured films. As the experimental results show in Figure 8b,c, the responses of the nanostructured film of ZnO/Ag (1 nm) under different gas concentrations are all below 20%, and the duration for the sensor to obtain a 90% reduction of its initial resistance requires more than 60 s in the gas concentration range of 10–250 ppm. Much better performances were obviously observed from the two nanostructured films of ZnO/Ag (3 nm) and ZnO/Ag (5 nm). It was also noticed that despite the larger value of S_max_ achieved from ZnO/Ag (3 nm) compared to the sample of ZnO/Ag (5 nm), ZnO/Ag (5 nm) performs better on the other parameters of S_60s_ and t_90%_, indicating an excellent performance in terms of the speed of response. As indicated in Appendix A, the total transversal area of macropore entrances in ZnO/Ag (3 nm) is slightly larger than the value in ZnO/Ag (5 nm), which can allow more gas to penetrate into the nanostructured film, achieving maximum sensitivity. For the other two parameters of S_60s_ and t_90%_, achieving a large entrance area of macropores is critical to the fast response achieved by the easy diffusion of ammonia molecules through the macropores to the inner surface of ZnO. It is believed that the gas diffusion of viscous flow and Knudsen flow mainly contribute to the fast response of gas sensing. The pore diameter is one of the most important factors for surface diffusion, and the effect of surface diffusion on gas mass transfer is negligible in large macropores (>50 nm) [36,37]. Thus, we calculated the total transversal area of pores above 1963.4954 nm^2^ (~50 nm), as shown in Appendix A. On the top surface, the total transversal areas of macropores with an area larger than 1963.4954 nm^2^ in ZnO/Ag (1, 3, 5 nm) were, respectively, 27,368.6753 nm^2^, 289,380.5096 nm^2^, and 303,789.3974 nm^2^. The total transversal area of macropore entrances on the top surface with an area larger than 1963.4954 nm^2^ in ZnO/Ag (5 nm) was slightly larger than the value from ZnO/Ag (3 nm). In addition, the round shape of the entrance to macropores shown in Appendix A could be more effective for the rapid diffusion of gas molecules than the stripped macropores with the same entrance area shown in Appendix A. Therefore, the sample of ZnO/Ag (5 nm) exhibits better performance on the parameters of S_60s_ and t_90%_.

Figure 8d shows the typical dynamic sensing spectrum of ZnO/Ag (5 nm) and pure ZnO sensors under the ammonia gas concentration of 300 ppm, and the process of a fast response upon inpouring the target gas can be clearly observed. As shown in Figure 8d, both the maximal response and response speed of ZnO are much lower than the latter. Meanwhile, it was also noticed that the recovery speed of the nanostructured ZnO film is much slower than its response speed, which can be attributed to the trapping of ammonia gas in the mesopores and macropores. That is to say, the ammonia molecules detached from the ZnO surface in the mesopores and macropores could be recaptured by the ZnO surface during their diffusion to the outside. Thus, the recovery of the nanostructured ZnO film is relatively slow. It should be noted that the data sampling interval during the characterization was 2 s.

As for the mechanism of the enhanced and fast response, the porous nanostructured film with an increased surface-to-volume ratio plays a crucial role in the achievement. Meanwhile, it is also necessary to discuss the role of silver, which is distributed around the nanostructured ZnO nanocrystals. To make a direct comparison, two samples of pure ZnO film and ZnO film with a 3 nm silver layer on the top surface were fabricated, which were thermally annealed under the same conditions. The SEM figure of ZnO with a 3 nm silver layer on top after thermal annealing is shown in Appendix A. The Ag particles clearly aggregated over the flat ZnO film surface, and no clear porous structure was observed, indicating that the formation of nanostructured film through the diffusion of silver atoms into ZnO film from the top is difficult. Furthermore, the response performance of thermally annealed ZnO film with silver on top was even lower than the pure ZnO film, as shown in Appendix A. Since the work function of silver (~4.26 eV) is slightly higher than the fermi level of n-type ZnO at room temperature, electrons will be transferred from silver to ZnO nanocrystals; the band diagram is shown in Appendix A. The additional free electrons on ZnO crystals may help to capture more O_2_ molecules from the background gas at room temperature [38]. However, when the temperature of sample is raised to 300 °C, a great amount of electron-hole pairs will be generated in ZnO crystals by thermal excitation [39], which induces the reduction of the fermi level of ZnO. Meanwhile, the free electrons generated from thermal excitation are helpful for capturing O_2_ from the background gas for the formation of O^2−^ on the ZnO surface [40], the contribution of which can be much larger than the electron donation from silver. As the band diagram in Appendix A shows, a thick depletion region will consequentially form on the ZnO surface, and the electrical resistance among ZnO grains will greatly increase. The resistance will be decreased upon the injection of ammonia gas into the chamber due to the dissociation of O^2−^ from the ZnO surface and the release of the free electrons. Therefore, based on the experimental result and theoretical analysis, the remaining Ag nanocrystals have a lower contribution to the ammonia gas sensing performance than thermal excitation under 300 °C heating conditions.

Generally, it is difficult to achieve high surface-to-volume ratios and superior sensing performance in compact films produced through the traditional physical vapor deposition approach. Our nanostructured film prepared through the physical method has been demonstrated to have better gas-sensing abilities than the traditional deposited film, as shown in Table 1. As a matter of fact, nanostructures composed by nanomaterials through chemical methods are superior due to their large surface-to-volume ratios. As shown in the Table 1, it can be observed that our work is no worse than these ZnO nanostructures. Therefore, our physical-vapor-deposited nanostructured films possessing mesopores and macropores have been demonstrated to achieve sensitive and rapid sensing performances.

## 4. Conclusions

The porous nanostructures of ZnO/Ag nanostructured film were experimentally prepared through physical vapor deposition of silver and ZnO layers, followed by thermal annealing in air. The diffusion of silver atoms around the ZnO nanograins and the induced film expansion during high temperature annealing can be regarded as the mechanism for the nanostructured film. Experimental results also demonstrate that the features of pores in the nanostructured film are dependent on the thickness of pre-deposited silver nano-layer. After the ammonia gas sensing characterization, all the nanostructured films pre-coated with different thicknesses of silver layers have been proven to exhibit enhanced gas-sensing abilities over the pure ZnO film, due to the increased effective surface area. As seen from the comparison study of the nanostructured films under gas concentrations of 10 ppm to 300 ppm at 300 °C, the sample of ZnO/Ag(3 nm) exhibits the highest maximum response value due to the largest effective surface areas of ZnO, while the sample of ZnO/Ag(5 nm) performs the quickest detection of ammonia gas due to by the larger size of macropores in the nanostructured film.

## Figures and Tables

**Figure 1 molecules-25-01899-f001:**
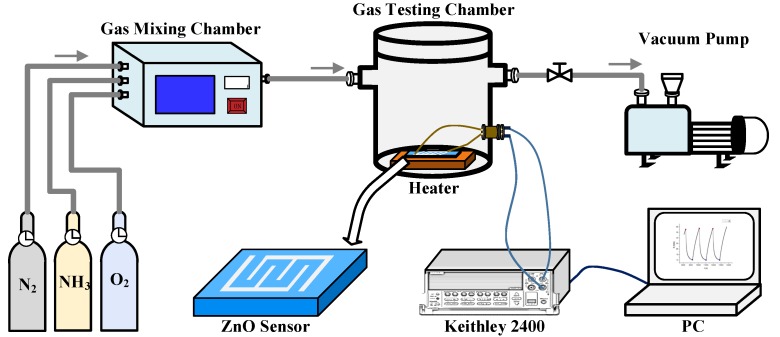
Diagram of the experimental system for NH_3_ gas sensing characterization.

**Figure 2 molecules-25-01899-f002:**
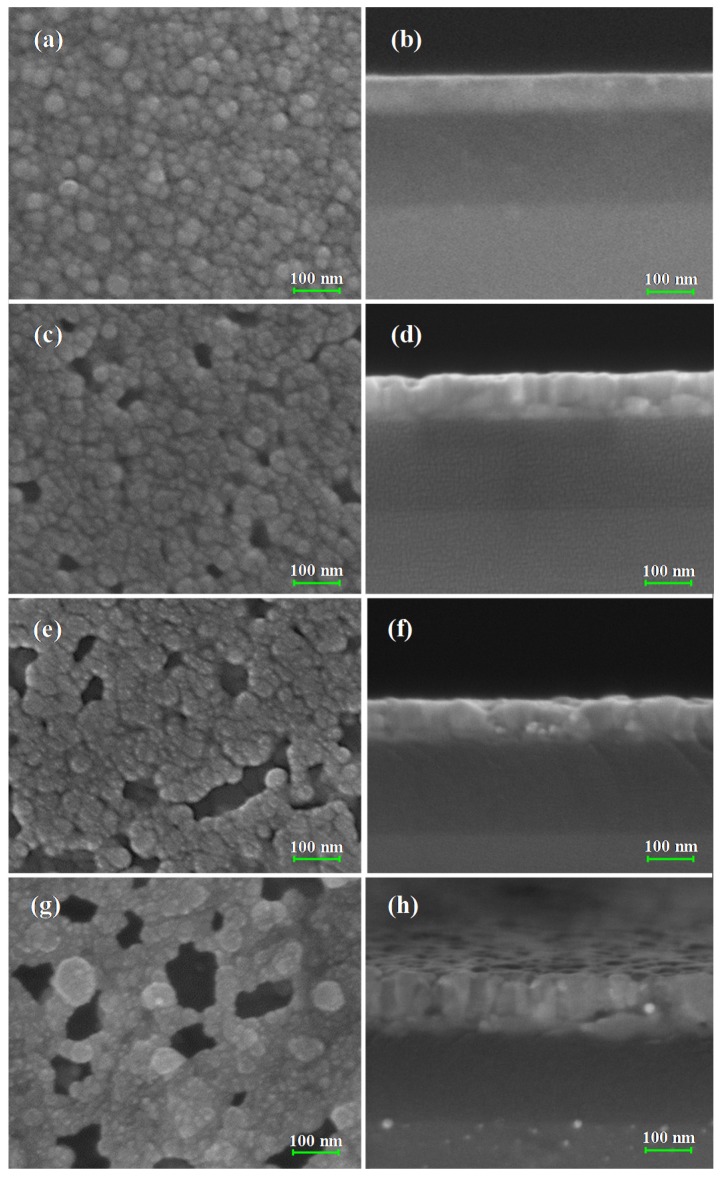
The SEM micrograph of ZnO films with different thicknesses of Ag: (**a**) top view, (**b**) side view of a 100 nm ZnO film without Ag; (**c**) top view, (**d**) side view of a 100 nm ZnO film with 1 nm of Ag; (**e**) top view, (**f**) side view of a 100 nm ZnO film with 3 nm of Ag; (**g**) top view, (**h**) side view of a 100 nm ZnO film with 5 nm of Ag.

**Figure 3 molecules-25-01899-f003:**
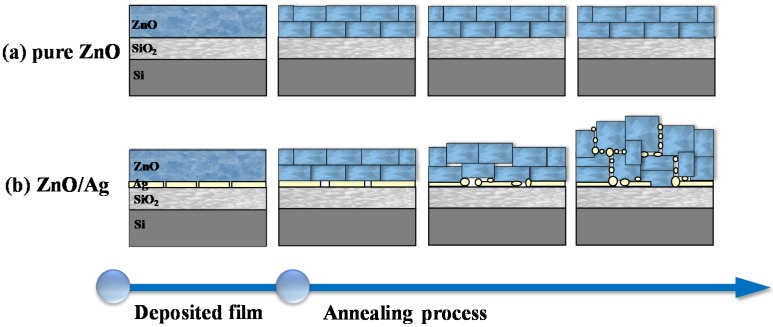
Schematic diagram of the morphological evolution of (**a**) pure ZnO film during thermal annealing and (**b**) ZnO/Ag nanostructured film during thermal annealing.

**Figure 4 molecules-25-01899-f004:**
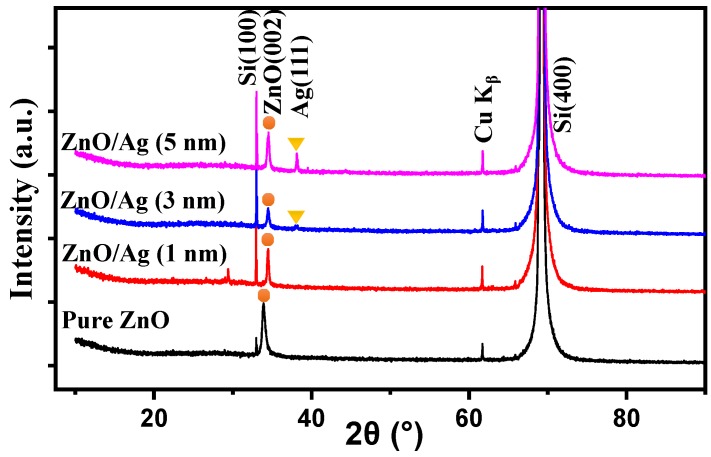
The XRD patterns of annealed ZnO films with different thicknesses of Ag layer.

**Figure 5 molecules-25-01899-f005:**
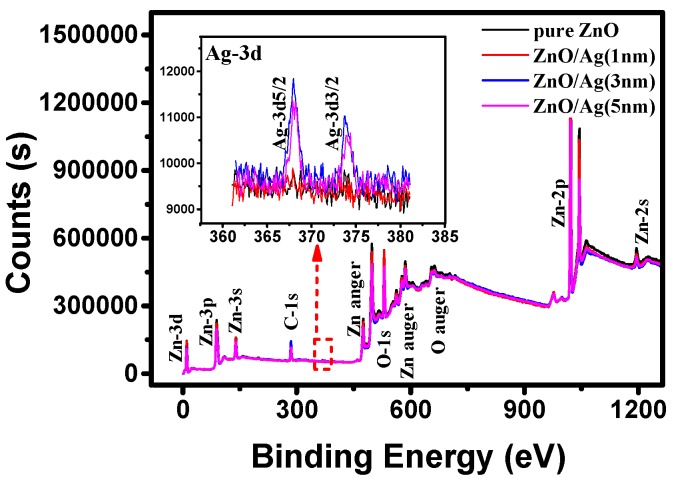
The XPS spectrum measured on the annealed ZnO films with different thicknesses of Ag layer.

**Figure 6 molecules-25-01899-f006:**
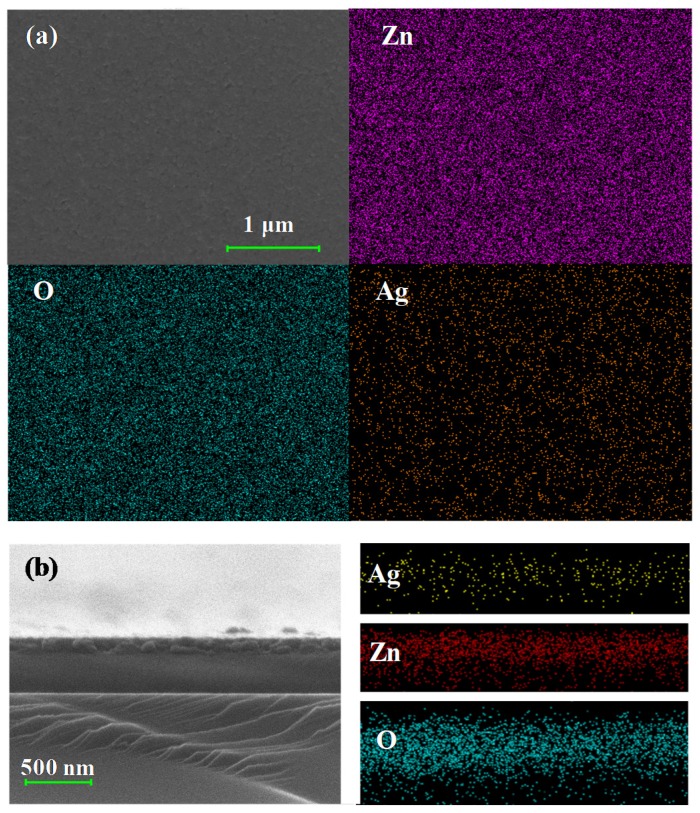
The EDS analysis of ZnO/Ag (5 nm) films: (**a**) the horizontal surface; (**b**) the cross-section.

**Figure 7 molecules-25-01899-f007:**
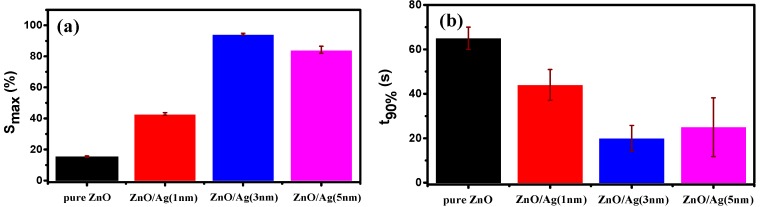
Sensing performance comparison of the annealed ZnO films with or without Ag layers: (**a**) gas response of ZnO films at 333 ppm of NH_3_; (**b**) response time for the resistance dropping 90% of the initial value.

**Figure 8 molecules-25-01899-f008:**
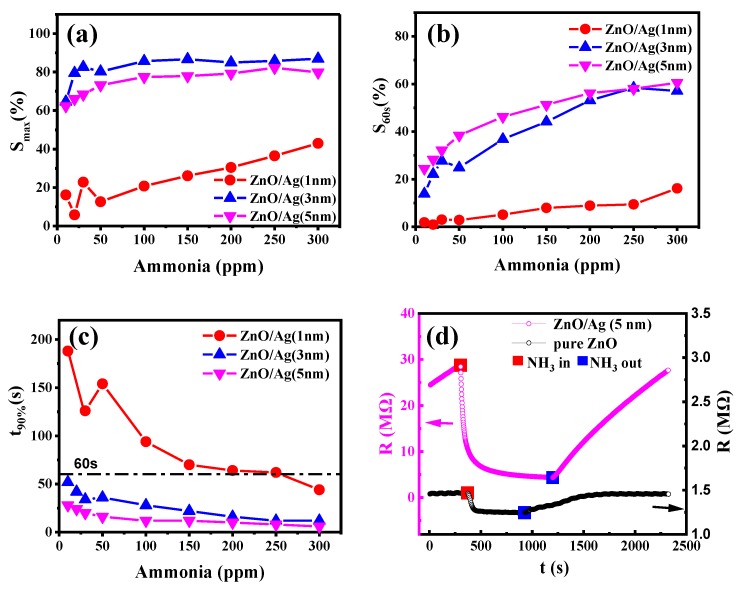
Sensing performance of nanostructured films with 1, 3, and 5 nm of Ag exposed to 10–300 ppm NH_3_: (**a**) maximum gas response of different Ag/ZnO films; (**b**) gas response of different Ag/ZnO films exposed to NH_3_ for 60 s; (**c**) response time for the resistance to drop to 90% of the initial value; (**d**) sensing performance of ZnO films with 5 nm of Ag exposed to 300 ppm NH_3_, compared to pure ZnO.

**Table 1 molecules-25-01899-t001:** The performance comparison of ZnO ammonia sensors.

	Material	Structure	NH_3_ Concentration	Response		Sensing Temperature
[15]	ZnO	traditional deposited film	600 ppm	160 s-57.5%	\	150 °C
[41]	ZnO	nanoparticles	800 ppm	48 s-99.61%	\	300 °C
[42]	ZnO	nanoplates	300 ppm	43 s-72.22%	\	300 °C
[43]	ZnO	nanoparticles	40 ppm	261 s-95.32%	\	350 °C
[44]	ZnO	nanofibers	20 ppm	~25%	\	550 °C
This work	Ag/ZnO	nanostructured film	300 ppm 10 ppm	60 s-61.18% 60 s-25.14%	6 s-10% 26 s-10%	300 °C

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
