# Peer review of "Nanostructured ZnO/Ag Film Prepared by Magnetron Sputtering Method for Fast Response of Ammonia Gas Detection"

_molecules, 2020, doi:10.3390/molecules25081899_

Round 1
Reviewer 1 Report
The paper under review describes fast response of ammonia gas detection fabricated by magnetron sputtering method technique. The work consists of some interesting techniques of film deposition and characterisation,in particular,deposition of porous electrodes by magnetron sputtering method. However, authors seem to have not given sufficient attention to preparation of manuscript. Authors must address a number of unclear points for the sake of clarity.
1.Why use Ag material and use 3 nm thickness or do not use other materials such as Pd film.
2. "The high-resolution XPS spectrum hows two obvious relative fewer amounts of silver atoms diffused from the bottom layer
How XPS and Ag is not oxidized after high temperature thermal annealing, is not mentioned.
The paper may be taken up for publication after minor revision
Author Response
Dear Reviewer:
We are resubmitting the enclosed manuscript entitled “Nanostructured ZnO/Ag film prepared by magnetron sputtering method for fast response of ammonia gas detection”(ID: Molecules-752830), which we wish to be considered for publication in “Molecules”.
The reviewers’ comments concerning our manuscript entitled Those comments are all valuable and very helpful for revising and improving our paper as well as the important guiding significance for our further research. Based on these comments and suggestions, careful revisions have been made on the original manuscript, in the hope that the issues raised may be duly addressed. Revised portion are highlighted in yellow in the re-uploaded paper. The major corrections in the paper and the responds to the reviewer’s comments are attached as following.
We appreciate the comments from you.
Thank you and best regards.
Yours sincerely,
Corresponding author: Haifei Lu
E-mall: haifeilv@whut.edu.cn
Department of Physics
Wuhan University of Technology
Physics Building
Wuhan 430070, China

Reviewer 2 Report
Report on « nanostructured ZnO/Ag film prepared by magnetron sputtering method for fast response of ammonia gas detection”
In this manuscript Y. Zheng et al. report on the elaboration of ZnO/Ag nanostructured sensor to detect with high accuracy and faster ammonia gas. They show that mesoporous film could be very efficient for this detection and that depositing an underlying Ag layer on SiO2 substrate could favor the formation of the mesoporous structure and thus increasing the surface/volume ratio favors the efficiency of the detection. Furthermore, they report data on the speed of the detection. The paper is well written and reports interesting results may be suitable for publication in Molecules, however, some changes or important precisions should be added before publication:
1-It is not obvious to compare with the state of the art, there are numerous works on similar ZnO based materials for ammonia detection. What is the real contribution of this mesoporous substrate compared to the classical deposition of nanoparticles with a very high surface/volume ratio?
2-As the formation of the mesoporous structure is highly dependent on the Ag underlying layer, what is the reproducibility of the structural organization and thus on the efficiency of detection.
3- The size pore appears to be very high (SEM) (I’m not convinced by the surfaces statistical analysis as the SEM side views show always very compact structures) and the access to the internal structure by gas diffusion is a limitation as observed to recovery the initial resistance affects gas exposition.
4-Authors claim that “the sensing ability of the porous nanostructured ZnO films, …, has been clearly demonstrated to be greatly enhanced”. That is true if they compare with pure ZnO compact films, but several teams report best results with a detection limit at 5 ppm of ammonia and there is no comparison for the speed of detection. So again, what the authors bring of very new?
Minor comments: the word ‘nanostructured’ is often spelled as ‘nanostrctured’
Author Response

(The authors gave the same response as above.)

Reviewer 3 Report
General comment:
The paper deals with experimental investigations on ZnO based materials design for ammonia detection. Materials characterization was performed using SEM, XRD and XPS techniques.
The manuscript is suitable to be published in this journal; however, some points should be addressed before publication. For example, it is not clear why no Ag is detected in ZnO/Ag(1nm) by any technique and no analysis is made to demonstrate that Ag thickness reach during magnetron sputtering is 1, 3 and 5 nm. In the same way, some differences between ZnO crystalline structures are discussed and no ZnO crystal size nor interplanar distance values are shown to support it.
The estimation of pore area using SEM images processed through Matlab provides values of transversal areas of pore entrance for a minimum part of the solid represented in microscopies. However, the area where ammonia interacts with the detector is the walls of the pores, unknown in the manuscript. The surface-to-volume ratio is the base for some conclusions and requires numeric values to support the discussion. The pore shape is also discussed to explain some results, but SEM images do not show information about pore tortuousness. A technique should be used to estimate the specific surface area of each prepared solid, as well as the pore size distribution and shape, and total pore volume, using more representative solid mass, N2 adsorption-desorption at 77 K, for example.
Some minor language mistakes are present that should anyway be corrected.
Introduction.
Please, include some data regarding the properties of ZnO to be used as NH3 detector.
Experimental details
2.2 Characterizations of ZnO sensors
Please, include conditions of XRD, SEM and XPS analysis.
Results and discussion.
Figure 2a and Figure S1a show no difference in materials structure, even when are from different materials.
Line 126- Please, use the established pore classification, the term “nanopore” is not appropriate to refer to the pore size in nanostructured materials.
Line 154- Please also correct the term “nano-pore”
Line 167- Please, justified why 4000nm2 is used as a comparative value between the solids pore area.
Line 170- Please, change the expression “We believe” and justified the assumption of the area and pores shapes are critical for surface-to-volume rate.
Figure 3. Please, identify the layers in scheme 'a' in a similar way as identified in the scheme 'b'.
Figure 4. Please, normalize the patterns and present the separate baselines to facilitate the comparison of the figure. Just for reference, I would like to suggest to consider (only as a visual guide) Figure 1 of the following manuscript:
Grams, J. et al. (2015) ‘Influence of Ni catalyst support on the product distribution of cellulose fast pyrolysis vapors upgrading’, Journal of Analytical and Applied Pyrolysis, 113, pp. 557–563. doi: 10.1016/j.jaap.2015.03.011
Figure 8d- Please, use a secondary y-axis.
Author Response

(The authors gave the same response as above.)

Round 2
Reviewer 2 Report
This new version is now acceptable for publication.
Author Response
Dear Reviewer:
We are resubmitting the enclosed manuscript entitled “Nanostructured ZnO/Ag film prepared by magnetron sputtering method for fast response of ammonia gas detection”(ID: Molecules-752830), which we wish to be considered for publication in “Molecules”.
The reviewer’ comments concerning our manuscript were all valuable and very helpful for revising and improving our paper as well as the important guiding significance for our further research.
We appreciate the comments from you.
Thank you and best regards.
Yours sincerely,
Corresponding author: Haifei Lu
E-mall: haifeilv@whut.edu.cn
Department of Physics
Wuhan University of Technology
Physics Building
Wuhan 430070, China
Reviewer 3 Report
The main results of ammonia detection capacity reported in the article are valuable and the authors solve the majority of the comments made before. However, the absence of morphological characterization must be resolved to be published. The absence of signal related to silver or silver compounds on ZnO/Ag (1 nm) in the XRD, XPS and EDS analyses must be explained as well.
Line 173- Please, justified the election of 4000 nm2 value (~70 nm of pore diameter) as the discriminating of pore size with an equation, a reference or experimental results, hence the ammonia kinetic diameter is less than one nm, quite smaller than pore diameter, and no mass transfer limitations were reported.
Line 177- Please, justified the pore shape with an appropriate result, it is not possible to assume the shape of the pores based on SEM images because the images do not represent a significant fraction of the solid, this conclusion must be corroborated with some morphological analysis.
Author Response
Dear Reviewer:
We are resubmitting the enclosed manuscript entitled “Nanostructured ZnO/Ag film prepared by magnetron sputtering method for fast response of ammonia gas detection”(ID: Molecules-752830), which we wish to be considered for publication in “Molecules”.
The reviewer’ comments concerning our manuscript entitled Those comments are all valuable and very helpful for revising and improving our paper as well as the important guiding significance for our further research. Based on these comments and suggestions, careful revisions have been made on the original manuscript, in the hope that the issues raised may be duly addressed. Revised portion are highlighted in yellow in the re-uploaded paper. The major corrections in the paper and the responds to the reviewer’s comments are attached as following.
We appreciate the comments from you.
Thank you and best regards.
Yours sincerely,
Corresponding author: Haifei Lu
E-mall: haifeilv@whut.edu.cn
Department of Physics
Wuhan University of Technology
Physics Building
Wuhan 430070, China
